# Comparative Study of the Nutritional Value and Degradation Characteristics of Amaranth Hay in the Rumen of Goats at Different Growth Stages

**DOI:** 10.3390/ani13010025

**Published:** 2022-12-21

**Authors:** Shengjun Zhao, Shilong Zhou, Yuanqi Zhao, Jun Yang, Liangkang Lv, Zibin Zheng, Honghua Lu, Ying Ren

**Affiliations:** 1Hubei Key Laboratory of Animal Nutrition and Feed Science, Wuhan Polytechnic University, Wuhan 430023, China; 2State Key Laboratory of Animal Nutrition, College of Animal Science and Technology, China Agricultural University, Beijing 100193, China; 3Fuxian Agricultural Technology Co., Ltd., Xiaogan 432800, China

**Keywords:** amaranth hay, growth stage, in vivo digestibility, goat

## Abstract

**Simple Summary:**

At present, studies on the rumen degradation characteristics of amaranth hay at different growth stages are limited. In the current study, four growth stages (squaring stage (SS), initial bloom stage (IS), full-bloom stage (FS) and mature stage (MS)) were selected. The chemical composition and rumen degradation characteristics of amaranth hay at four stages were studied in detail. Among the four stages, IS was superior in terms of chemical composition and rumen degradability characteristics.

**Abstract:**

The objective of this study was to investigate the rumen degradation characteristics of grain amaranth hay (*Amaranthus hypochondriacus*) at four different growth stages. The aim of this study was to evaluate the nutritional value of grain amaranth hay at different growth stages by chemical composition, in vivo digestibility, and in situ degradability. Three Boer goats with permanent ruminal fistulas were selected in this study. Amaranthus hay at four different growth stages (squaring stage (SS), initial bloom stage (IS), full-bloom stage (FS) and mature stage (MS)) was crushed and placed into nylon bags. Each sample was set up with three replicates, and two parallel samples were set up in fistulas at each time point. The rumen degradation rates of dry matter (DM), crude protein (CP), neutral detergent fibre (NDF) and acid detergent fibre (ADF) were determined at 0, 6, 12, 24, 36, 48 and 72 h. The results were as follows: (1) The concentration of CP in SS was the highest and was significantly higher than that in other stages (*p* < 0.05), whereas the contents of NDF and ADF gradually increased with the extension of the growing period and reached a maximum in MS; (2)The degradation of CP in the rumen at 72 h of SS and IS was more than 80%. Compared with other stages, the effective degradability of CP was highest in SS (*p* < 0.05) and reached 87.05% at 72 h, and the degradation rate was the lowest in MS; and (3) The effective degradability of NDF in IS was the highest (*p* < 0.05) and reached 69.326% at 72 h. The effective degradability of ADF in MS was the highest (*p* < 0.05) and reached 65.728% at 72 h. The effective degradability of DM and CP in SS was the highest. In conclusion, among the four stages, IS was superior in chemical composition and rumen degradability characteristics.

## 1. Introduction

*Amaranthus hypochondriacus* is an annual herbaceous plant of amaranth that originated from tropical and subtropical areas of Central America and Southeast Asia. Amaranthus hypochondriacus has the advantages of strong adaptability, fast growth, wide planting range and high yield, and can grow in areas lacking water and those with poor soil. Amaranth seeds are rich in nutrients and contain many essential amino acids such as lysine and methionine [1]. Amaranth can be used to replace some of the corn, especially in arid regions [2]. Ensiled amaranth has a high yield of dry matter (DM) and crude protein (CP), and in vitro methane production, which could be a potential feed resource for ruminants [3]. Amaranth also has abundant resources in China, and its annual planting area is more than one million hectares. In northern China, the yields of fresh grass and hay can reach 130 t/hm^2^ and 20 t/hm^2^, at the mature stage, and the crude protein of amaranth can reach 14% at the squaring stage [4]. The crude protein content of amaranth stems and leaves is very high, approximately 16–23% under dry matter conditions, especially the lysine content, which is very high, approximately 1%. In addition, the cp content of amaranth was higher than corn and reached 285 g/kg, and the lignin concentration was lower than corn and reached 40 g/kg [5]. Supplementing grain amaranth in the diet has no adverse effects on the health and performance of dairy cows, and can improve the economic efficiency of the pasture [4]. Therefore, it can be regarded as a kind of feed resource and has good prospects for development and application.

Studies have shown that the replacement of animal protein meat and bone meal in broiler diets with untreated and heat-treated grain amaranth meal has no significant effect on the growth performance of broilers [6]. The use of different proportions of amaranth silage instead of corn silage has no effect on the performance of dairy cows and can be used as high-quality feed for dairy cows [2]. When grain amaranth silage was used to replace corn silage in lambs’ diet, it was found that with the increase of grain amaranth silage substitution ratio, lambs’ feed intake and daily gain were increased, but the effect on digestive ability was not significant [7]. Rahjerdi et al. studied the potential of two Amaranthus varieties, maize and the combination of Amaranthus and maize as feed sources for ruminants, and measured their chemical composition, silage fermentation characteristics, rumen degradation rate of DM and digestibility in sheep [8].

The segment of the growth stage is one of the most crucial aspects of the herding grass quality. Plant maturity was found to affect NDF, ADF, CP, lignin and nitrate [9]. With the increase in plant maturity, the contents of NDF, ADF and lignin in straw increased, while the contents of crude protein and soluble carbohydrate compounds decreased [10]. When the forage was harvested too early, the yield was low, and the moisture content was high. However, nutrient content and digestibility decreased when the forage was harvested late. It is well known that lucerne and other fodder crops generally lose some of their feeding value as the plant matures [11]. As a result, the right growing stage has a significant impact on the forage’s quality. In addition, the nutrients degradation in the rumen is a crucial indicator of the ruminant feed nutritional value evaluation. The degradation rates of DM, CP, NDF and ADF in the rumen are crucial indices to determine the quality of ruminant feed [12]. At present, the methods used to evaluate the degradation rate of ruminant feed are mainly in vivo, in situ and in vitro. The half in vivo method refers to the nylon bag method, which is simple and reproducible, can truly reflect the internal environment of the rumen, and is convenient for batch operation. It is one of the international methods for determining the rumen degradation rate of feed. The effects of amaranth at different growth stages on the rumen degradation rate of goats were not clear. Therefore, the purpose of this study was to evaluate the changes in rumen degradation characteristics during the growth of *Amaranthus hypochondriacus*.

## 2. Materials and Methods

### 2.1. Animal Care

The animal trial was conducted according to the Animal Scientific Procedures Act 1986 (Home Office Code of Practice. HMSO: London January 1997) and EU regulations (Directive 2010/63/EU). This experimental protocol used in this study was approved by the Institutional Animal Care and Use Committee of Wuhan Polytechnic University. (Hubei, China; approval number: 2010-0029).

### 2.2. Preparation of Amaranth

Amaranth was provided by Fuxian Agricultural Technology Co., Ltd. (Xiaogan, China), and grown in 2021 in Guangzhou, Guangdong province in China (23°30′ N and 113°83′ E). Amaranth was cut at four growth stages (SS = squaring stage; IS = initial bloom stage; FS = full-bloom stage; MS = mature stage). The whole-plant amaranth was cut short into 4 cm-long pieces. Samples of fresh forages were dried in a forced-air oven at 65 °C for two days and then removed for 1 day to make air-dried samples. Four different growth stages of amaranth were smashed by a grinder, and the smashed samples used for nutrient composition analysis were passed through a 1-mm sieve, while the smashed samples for the rumen degradation study were passed through a 2-mm sieve.

### 2.3. Animals and Diets

Three Boer goats (body weight: 30.7 ± 1.2 kg) fitted with a rumen fistula for measuring rumen degradation characteristics were housed in the laboratory animal room at Wuhan Polytechnic University (China). These goats were fed 40:60 concentrate to roughage ration at 08:00 and 16:00. Water was always available. The diets were prepared according to NY/T816-2004 meat sheep feeding standards, and the nutrient levels of the basal diet are shown in Table 1.

### 2.4. In Situ Ruminal Incubation

The size of the nylon bags was 10 cm × 6 cm and the aperture was 48 µm (300 mesh). The nylon bags treated in the rumen were removed, washed repeatedly, dried in a 65 °C oven for two days, placed for one day, and weighed. Each sample was carefully weighed at three grams and put into a nylon bag. The samples were kept in a nylon bag that was fastened to a flexible semi-plastic hose. Each sample was divided into three parts and placed into the rumens of three goats, with two parallels set for the same fistulated goat at each time point. The “gradual addition/all out” protocol was used for the rumen incubations. Incubations were carried out for 72, 48, 36, 24, 12, and 6 h; bags were inserted at 20:00 (D 1), 20:00 (D 2), 08:00 (D 3), 20:00 (D 3), 08:00 (D 4), and 14:00 (D 4), and all were removed at 20:00 (D4). The nylon bags were taken from the rumen after incubation and rinsed until the rinse water was clear. This was done to eliminate extra ruminal contents and bacteria on the surface and to stop microbial activity. The cleaned nylon bags and the residues were subsequently dried at 65 ℃ to a constant weight. Only washing was performed on the 0 h incubation samples under the identical circumstances. The residues were weighed and ground through a 1-mm sieve and mixed and analyzed for nutrient content (DM, CP, NDF, and ADF).

### 2.5. Rumen Degradation Models

Yielding estimates for the percentages of ruminal DM, CP, NDF, and ADF degradability (P) at time (t) were based on an exponential curve as P = A + B (1 − e^−Ct^), and an iterative regression analysis was used to fit the date [13]. The effective degradability (ED) values of DM, CP, NDF, and ADF were then calculated as ED (%) = A + B × C/(C + k), according to Ørskov et al. In these equations, “e” is the base of natural logarithms, “A” is the soluble and very rapidly degradable fraction; “B” is the insoluble but potentially degradable fraction that degrades at a constant fractional rate (C) per unit time (t), ED is the effective degradability, and “k” refers to the fractional outflow rate from the rumen. An assumed value for “k” was 3.1%/h [14]. The values for a, b and c were calculated by nonlinear regression with SPSS software.

### 2.6. Chemical Analysis

Dry matter (DM) was calculated by drying to a constant weight at 105 °C. Kjeldahl digestion and distillation was used to analyze the nitrogen (AOAC 1990) [15].

Dry matter (DM) was determined by drying at 105 °C to constant weight. Nitrogen was analyzed by Kjeldahl digestion and distillation (AOAC 1990), and CP was obtained as N × 6.25 [15]. The contents of NDF and ADF in the samples were analyzed according to Van Soest et al. [16].

### 2.7. Statistical Analysis

Data were analyzed by one-way ANOVA with SPSS statistical software (Ver. 20.0 for Windows; SPSS, Chicago, IL, USA). The statistically significant differences were determined by Duncan’s multiple range tests. Data are presented as the mean and SEM. The significance level was indicated at *p* < 0.05, with trends declared at 0.05 < *p* < 0.10.

## 3. Results

### 3.1. Chemical Composition

As shown in Table 2, the nutrient composition of amaranth hay was affected by different growth stages. When the development stage was extended, the DM contents gradually increased, and the concentrations of NDF and ADF showed similar trends. The concentration of DM in MS was substantially greater than those in the other stages (*p* < 0.05), and gradually increased when the growth stage was extended. The highest CP content was found in SS (*p* < 0.05).

### 3.2. Ruminal DM Degradation

Table 3 shows the rumen degradability and degradation parameters of DM. The ruminal DM degradation in SS was higher at 72 h compared to other stages (*p* < 0.05). Before 24 h, all of the amaranth hay degraded more quickly before leveling off. In comparison to the other stages, FS’s rapidly degradable fraction was largest (*p* < 0.05). In comparison to the other stages, the potentially degradable fraction of IS was highest (*p* < 0.05), but was not significantly different from that of SS (*p* > 0.05). The effective degradability of amaranth steadily decreased with the lengthening of the growing time, and the differences were significant (*p* < 0.05).

### 3.3. Ruminal CP Degradation

Table 4 shows the rumen degradability and degradation parameters of CP. In comparison to the other stages, the ruminal CP degradation at 72 h in SS was highest (*p* < 0.05), reaching approximately 87%. The rapidly degradable fraction and effective degradability of SS were significantly higher than those of the other stages (*p* < 0.05). Otherwise, the slowly degradable fraction of IS was largest and was similar to the ruminal DM degradation.

### 3.4. Ruminal NDF Degradation

Table 5 shows the rumen degradability and degradation parameters of NDF. In comparison to the other stages, the rumen degradability at 72 h and effective degradability of NDF in IS were highest (*p* < 0.05). Compared with MS, the rumen degradability of NDF at 72 h in FS was not noticeably different (*p* > 0.05). In comparison to the other stages, the rapidly degradable fraction of FS and the potentially degradable fraction of SS was highest (*p* < 0.05).

### 3.5. Ruminal ADF Degradation

Table 6 shows the rumen degradability and degradation parameters of ADF. In comparison to the other stages, the rumen degradability at 72 h and effective degradability in the MS of ADF were highest. We found that the rapidly degradable fraction of SS was slow, and the maximum was only 6.599%. Similar to the ruminal DM degradation, in comparison to the other stages, the rapidly degradable fraction of FS, and the potentially degradable fraction of IS was highest (*p* < 0.05).

## 4. Discussion

### 4.1. Chemical Composition

The primary factor impacting forage quality is its nutritional composition, and the forage quality is also affected by many other factors, such as variety, growing environment, climate and growing period [17,18]. With increasing growth time, the contents of dry matter and organic matter increased, the content of crude protein decreased gradually, the contents of structural carbohydrate and water complexes eventually caused CP to decline and NDF and ADF to rise, the overall nutritional value showed a downwards trend [9], and the palatability decreased. With the lengthening of the growing season, the contents of DM, NDF and ADF in grain amaranth feed gradually increased, while the content of CP gradually decreased [17,18]. Sarmadi et al. [19] showed that the CP content in mature amaranth was significantly lower than that in the initial bloom stage, which agreed with the findings of this study. Thornton [20] also found that the degradation rate of CP was closely related to the growth period. In this study, the nutrient composition of amaranth in different growth stages was constantly changing, and the contents of NDF and ADF increased continuously with the progression of the growth stages, while CP showed an inverse trend. The reason for these trends may be that in the process of transition from vegetative growth to reproductive growth, amaranth plants need to consume large amounts of nutrients to meet the development of their reproductive organs, leading to increasing fibre components in the cell walls of plants, decreases in the composition of the cell contents, a greater lignification degree, a gradual increase in the ratio of stem to leaf, and a decrease in the proportion of leaves in the whole plant’s structural carbon combined with the water content, finally leading to a decrease in the CP equal content, leading to decreasing the nutrient content of the herbage. The fibrous substances in straw include NDF, ADF and lignin, among which NDF and ADF are of nutritional significance to herbivorous domestic animals. NDF is the most effective index to reflect the fibre quality of feed, and it is also an important index to reflect the concentrate to roughage ratio of ruminants, while ADF can reflect the digestibility rate of roughage. Ma [21] showed that the contents of NDF (54.14%) and ADF (38.03%) of the amaranth reached their highest values in HS, and we found that the contents of NDF (64.92%) and ADF (51.80%) reached their highest values in MS in this study.

### 4.2. Ruminal DM Degradation

The degradation rate of dry matter in the rumen is an important factor affecting DM intake in ruminants, and its value is influenced by the content of fibre and lignification degree of the feed materials. The degradation rate of DM of roughage in the rumen increased with the extension of degradation time, but the rates of roughage degradation of samples of different qualities were different. The rumen degradation characteristics of DM in amaranth were different at different growth stages, and the 72 h degradation rate and effective degradation rate of DM decreased with the development of growth stages. In this study, the DM degradation rate of amaranth hay in different growth stages decreased with the lengthening of rumen retention time. In this study, the rapidly degradable fractions in different growth stages of amaranth ranged from 36.67% to 43.36%, and the potentially degradable fractions ranged from 27.29% to 38.05%. The highest dry matter degradability of alfalfa was 77.23%. The DM degradability of different types of maize was the highest in whole maize silage (64.89%), followed by corn stalk silage and corn stalk after 72 h of incubation in the rumen [8]. Ma et al. [22] demonstrated that the degradation rate of DM in cows’ rumens at SS of amaranth was higher than those in the initial bloom, full-bloom and mature stages, and the results of this study were similar. In comparison to the other stages, the effective degradability of hay DM in SS was significantly increased. In this study, the highest dry matter degradation rates were 67.98% in SS and 58.06% in MS, which were higher than those in whole corn silage and could be used as a substitute to a certain extent. The DM effective degradability of amaranth hay decreased with the extension of the growth period, which was consistent with Sarwar’s study [23]. Studies have shown that the DM effective degradability of amaranth hay in the squaring, initial bloom, full-bloom and mature stages were 56.71%, 51.29%, 50.73% and 50.57%, respectively, which were all lower than the results in the corresponding growth stage in this study, which may be caused by the variety of amaranth, the planting area, sample treatment factors and the experimental animals used [24]. For DM degradability, a decrease in CP and an increase in the fiber components were related to the declining fractions “A” and “B” for DM degradability as the plant continued in the growth stage, which led to a decline in the overall degradable DM fractions, or “A + B” [19]. The decreasing “C” with advancing age of the amaranth was consistent with that found for the mottgrass [23].

### 4.3. Ruminal CP Degradation

CP is one of the important indices to evaluate the quality of roughage, and its degradation in the rumen is influenced by the content of true protein, retention time in the rumen and degree of fermentation; therefore, it is important to study the degradation characteristics of CP in roughage [25]. In this study, the 72 h degradation rate and rumen effective degradability of CP in amaranth hay at SS were the highest, at 87.05% and 75.81%, respectively, and the rapid degradation portion was the highest, indicating that CP in amaranth hay at SS was easier to degrade. Yao Qing [26] showed that the larger the content of CP in the feed, the greater the rumen degradation rate, which agreed with the results of this study. Protein in the feed can be divided into three parts: the rapidly degraded fraction, the potentially degradable fraction and the degradation rate of the slowly degraded fraction. The proportions of each part in different feeds are different. The values of a, b, and c in the regression formula represent the contents of these three parts. The large differences in fraction “A” that existed between SS and MS were primarily caused by differences in N fractions within leaf tissue. Compared with SS, IS had a lower concentration of fraction “A” and a larger concentration of fraction “B” of CP, which could be caused by the increased concentration of CP bonded to cell wall components. [27]. Cell walls are a major part of the plant [28]. The amount and composition of the cell walls are probably the most important factors in determining the nutritive value of the plant [29]. As amaranth grows, the lignification of plants becomes increasingly serious, the increasing lignification degree affects the release and decomposition of nitrogen, nutrients are gradually transferred to seeds, and the CP content and biodegradable parts are gradually reduced, leading to a decrease in the CP degradation rate. Satter [30] found that the rumen degradation rate of CP was influenced by the nature of the feed itself, and the rapidly degraded fraction and the potentially degradable fraction in different feeds were different. In comparison to SS, the higher fraction “B” in IS for DM degradability may be linked to its higher CP concentration [31]. The amaranth in this study exhibited a lower ED of DM in MS, which is likely because of its increased ADL content, and high ADL concentrations can inhibit ruminal degradability. In this study, in comparison to SS, amaranth in IS contained less fraction “A” and more fraction “B”, which could be due to the fact that more CP is attached to the components of the cell wall.

### 4.4. Ruminal NDF and ADF Degradation

The NDF and ADF of roughage are the most difficult parts of feed to digest, and play important roles in maintaining ruminant digestion and absorption and rumen health. A higher fibre degradation rate is beneficial to the high concentration of volatile fatty acids produced by fibre fermentation, which can provide sufficient energy for animals and improve animal performance [21]. The decrease in rumen degradability of NDF with plant maturity is likely due to the increased content of lignin [17]. Lignin, because of its characteristic phenolic component, cannot be digested under anaerobic conditions and can reduce the proportion of potentially digestible fibre in herbage [32]. In this study, the change trends of the rumen degradation rates of NDF and ADF were different with the extension of the rumen degradation time, which was mainly caused by the differences in NDF and ADF components. Rosser [27] found that with the maturation of whole crops, NDF and ADF gradually increased. Different fibre compositions will affect the rumen degradation rate, so NDF and ADF at different growth stages have different degradation rates in the rumen [33]. NDF and lignin increase with plant aging, but CP and water-soluble carbohydrates decline and digestibility decreases. Studies have shown that the rumen degradability of NDF is negatively correlated with the content of NDF [34]. The effective NDF degradation rate of the hay in IS was significantly larger than those in the other three stages, which is because of the larger digestible nutrient content of the plants in IS. The rumen degradability and effective degradability of Amaranth in IS were the highest and were significantly higher than those in FS and MS. A possible reason is that amaranth in this stage has become yellow and lignified, which may potentially reduce the effective degradation rate of NDF. In general, ADF is the most difficult part of the feed to degrade. In this study, the effective degradability of ADF in amaranth hay in MS was the highest, which was consistent with the research results of Sun Guoqing [35]. In Sun Guoqing’s study, the ADF content of the amaranth in the mature stage was 32.40%, and in this study, it was 54.50%, which was higher, most likely because we used silage.

## 5. Conclusions

With the extension of the growing period, the contents of DM, NDF and ADF of grain amaranth feed gradually increased, while the content of CP gradually decreased. Among the four stages, IS was superior in chemical composition and rumen degradability characteristics.

## Figures and Tables

**Table 1 animals-13-00025-t001:** Dietary formula for fistulated goats (air-dried basis).

Ingredients	Proportion (%)	Nutrient Levels	
Peanut seedling	60.00	Metabolizable energy (MJ/kg)	8.78
Maize	15.20	Crude protein	15.03
Soybean meal	14.20	Neutral detergent fibre	41.07
Wheat bran	7.00	Acid detergent fibre	34.99
Soda	0.80	Calcium	0.66
NaCl	0.60	Calculated analysis	0.22
Premix	2.00		
Probiotics	0.20		
Total	100.00		

Note: (1) Per kilogram premix contains Vitamin A 560,000 IU, Vitamin D3 135,000 IU, Vitamin E 3,600,000 IU, Zinc 3100 mg, Magnesium 1850 mg, Manganese 1450 mg, Copper 1350 mg, Selenium 40 mg, Iodine 83 mg, and Cobalt 55 mg. (2) The metabolizable energy of the nutrient level was calculated, and other values were measured.

**Table 2 animals-13-00025-t002:** Nutrient compositions of *Amaranthus hypochondriacus* hay in different growth stages.

Items	SS	IS	FS	MS	SEM	*p*-Value
DM (%)	91.347 ^c^	91.611 ^c^	93.369 ^b^	94.095 ^a^	0.143	<0.001
CP (%)	19.985 ^a^	16.157 ^b^	15.018 ^b^	13.982 ^b^	1.425	0.014
NDF (%)	56.897	63.407	64.128	64.920	2.774	0.069
ADF (%)	29.558 ^d^	36.968 ^c^	43.621 ^b^	51.802 ^a^	2.426	<0.001

Values are presented as the means. SS = squaring stage; IS = initial bloom stage; FS = full-bloom stage; MS = mature stage. DM = dry matter; CP = crude protein; NDF = neutral detergent fibre; ADF = acid detergent fibre. In the same row, different letters on the data shoulder marks indicate significant differences (*p* < 0.05).

**Table 3 animals-13-00025-t003:** Rumen DM degradation characteristics of *Amaranthus hypochondriacus* hay.

Item	SS	IS	FS	MS	SEM	*p*-Value
Rumen Degradation Rate (%)						
0 h	42.170 ^a^	41.728 ^a^	43.365 ^a^	37.233 ^b^	0.802	<0.001
6 h	53.781 ^a^	45.847 ^b^	52.315 ^a^	45.125 ^b^	1.051	<0.001
12 h	65.492 ^a^	58.983 ^b^	56.257 ^bc^	53.864 ^c^	2.080	0.003
24 h	73.793 ^a^	73.432 ^a^	65.651 ^b^	62.522 ^c^	0.517	<0.001
36 h	75.342 ^a^	73.505 ^b^	66.665 ^c^	64.191 ^d^	0.742	<0.001
48 h	76.766 ^a^	74.449 ^b^	68.534 ^c^	67.241 ^c^	0.625	<0.001
72 h	78.194 ^a^	75.277 ^b^	70.734 ^c^	68.941 ^d^	0.357	<0.001
Rumen Degradation Parameters						
A/%	41.524 ^b^	39.04 ^c^	43.360 ^a^	36.676 ^d^	0.782	<0.001
B/%	36.461 ^a^	38.052 ^a^	27.295 ^c^	32.511 ^b^	1.105	<0.001
C/(%/h)	0.082 ^a^	0.063 ^b^	0.062 ^b^	0.060 ^b^	0.008	0.065
A + B/%	77.985 ^a^	77.092 ^a^	70.655 ^b^	69.186 ^c^	0.616	<0.001
ED/%	67.984 ^a^	64.584 ^b^	61.405 ^c^	58.063 ^d^	0.432	<0.001

A = rapidly degraded part of the grain amaranth hay sample, B = slowly degraded part, C = degradation rate of the slowly degraded part of B, A + B = potentially degradable part, ED = effective degradability. In the same row, different letters on the data shoulder marks indicate significant differences (*p* < 0.05).

**Table 4 animals-13-00025-t004:** Ruminal CP degradation characteristics of *Amaranthus hypochondriacus* hay.

Item	SS	IS	FS	MS	SEM	*p*-Value
Rumen Degradation Rate (%)						
0 h	54.745 ^a^	45.227 ^c^	50.958 ^b^	35.840 ^d^	0.914	<0.001
6 h	64.259 ^a^	55.845 ^c^	59.258 ^b^	44.084 ^d^	1.299	<0.001
12 h	73.172 ^a^	62.810 ^b^	64.325 ^b^	51.769 ^c^	1.995	<0.001
24 h	77.553 ^a^	77.799 ^a^	69.530 ^b^	54.557 ^c^	0.485	<0.001
36 h	80.696 ^a^	78.627 ^a^	70.975 ^b^	62.647 ^c^	0.931	<0.001
48 h	86.380 ^a^	79.712 ^b^	71.888 ^c^	65.720 ^d^	0.431	<0.001
72 h	87.050 ^a^	83.318 ^b^	74.564 ^c^	68.934 ^d^	0.463	<0.001
Rumen Degradation Parameters						
A/%	55.275 ^a^	44.550 ^c^	51.109 ^b^	36.690 ^d^	1.345	<0.001
B/%	31.877 ^b^	38.831 ^a^	22.463 ^c^	34.134 ^b^	1.639	<0.001
C/(%/h)	0.056 ^ab^	0.061 ^a^	0.074 ^a^	0.039 ^b^	0.009	0.029
A + B/%	87.152 ^a^	83.381 ^b^	73.572 ^c^	70.824 ^d^	0.546	<0.001
ED/%	75.813 ^a^	70.280 ^b^	66.757 ^c^	55.627 ^d^	0.445	<0.001

A = rapidly degraded part of the grain amaranth hay sample, B = slowly degraded part, C = degradation rate of the slowly degraded part of B, A + B = potentially degradable part, ED = effective degradability. In the same row, different letters on the data shoulder marks indicate significant differences (*p* < 0.05).

**Table 5 animals-13-00025-t005:** Rumen NDF degradation characteristics of *Amaranthus hypochondriacus* hay.

Item	SS	IS	FS	MS	SEM	*p* Value
Rumen Degradation Rate (%)						
0 h	32.673 ^b^	38.485 ^a^	39.25 ^a^	33.998 ^b^	1.279	0.002
6 h	35.846 ^c^	44.023 ^a^	43.194 ^a^	38.326 ^b^	1.057	<0.001
12 h	49.461 ^ab^	53.033 ^a^	45.611 ^b^	46.249 ^b^	2.543	0.066
24 h	61.665 ^b^	66.473 ^a^	55.026 ^c^	54.118 ^c^	1.358	<0.001
36 h	62.992 ^b^	66.892 ^a^	57.753 ^c^	54.575 ^d^	0.503	<0.001
48 h	64.360 ^b^	67.572 ^a^	58.392 ^c^	59.968 ^c^	0.883	<0.001
72 h	67.072 ^b^	69.326 ^a^	62.557 ^c^	60.673 ^c^	0.836	<0.001
Rumen Degradation Parameters						
A/%	30.201 ^c^	36.712 ^a^	38.467 ^a^	33.035 ^b^	0.777	<0.001
B/%	37.972 ^a^	33.783 ^b^	26.491 ^c^	28.755 ^c^	1.081	<0.001
C/(%/h)	0.056 ^a^	0.061 ^a^	0.036 ^b^	0.049 ^ab^	0.007	0.049
A + B/%	68.172 ^a^	70.495 ^a^	64.958 ^b^	61.790 ^c^	1.261	0.001
ED/%	54.585 ^b^	59.078 ^a^	52.334 ^c^	50.525 ^d^	0.493	<0.001

A = rapidly degraded part of the grain amaranth hay sample, B = slowly degraded part, C = degradation rate of the slowly degraded part of B, A + B = potentially degradable part, ED = effective degradability. In the same row, different letters on the data shoulder marks indicate significant differences (*p* < 0.05).

**Table 6 animals-13-00025-t006:** Rumen ADF degradation characteristics of *Amaranthus hypochondriacus* hay.

Items	SS	IS	FS	MS	SEM	*p*-Value
Rumen Degradation Rate (%)						
0 h	6.488 ^c^	10.114 ^b^	20.098 ^a^	18.786 ^a^	1.070	<0.001
6 h	25.614 ^d^	36.295 ^c^	42.637 ^b^	44.958 ^a^	0.796	<0.001
12 h	36.406 ^c^	43.976 ^b^	46.505 ^b^	51.220 ^a^	1.926	<0.001
24 h	47.764 ^b^	57.377 ^a^	49.044 ^b^	58.199 ^a^	0.897	<0.001
36 h	48.118 ^b^	57.696 ^a^	49.704 ^b^	60.605 ^a^	1.387	<0.001
48 h	50.414 ^c^	59.216 ^b^	51.776 ^c^	65.699 ^a^	0.949	<0.001
72 h	53.860 ^d^	60.938 ^b^	56.960 ^c^	65.728 ^a^	0.357	<0.001
Rumen Degradation Parameters						
A/%	6.599 ^c^	10.638 ^b^	20.415 ^a^	19.775 ^a^	1.449	<0.001
B/%	45.393 ^b^	49.205 ^a^	31.49 ^c^	43.811 ^b^	1.513	<0.001
C/(%/h)	0.089 ^b^	0.108 ^b^	0.177 ^a^	0.118 ^b^	0.015	0.002
A + B/%	51.992 ^c^	59.843 ^b^	51.905 ^c^	63.586 ^a^	0.856	<0.001
ED/%	40.265 ^d^	48.869 ^b^	47.212 ^c^	54.409 ^a^	0.465	<0.001

A = rapidly degraded part of the grain amaranth hay sample, B = slowly degraded part, C = degradation rate of the slowly degraded part of B, A + B = potentially degradable part, ED = effective degradability. In the same row, different letters on the data shoulder marks indicate significant differences (*p* < 0.05).

## Data Availability

The data used and analyzed in the current study are available from the corresponding author on reasonable request.

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
