# Peer review of "Comparative Study of the Nutritional Value and Degradation Characteristics of Amaranth Hay in the Rumen of Goats at Different Growth Stages"

_animals, 2022, doi:10.3390/ani13010025_

Round 1
Reviewer 1 Report
Comments to the Author
Major concerns
The manuscript as I believe investigated the the nutritional value and degradation characteristics of amaranth hay in the rumen of goats at different growth stages. It’s a valuable work and worthy to be done. This is a good help for livestock producers to better use the forage material-amaranth hay. While this may be of interest I found the the manuscript presents in not good form.
Question 1: The form of the Abstract, it doesn't seem to do a good job of presenting the conclusions of the study briefly.
Question 2: In the Introduction, author used too much space to introduce the basic information of Amaranthus hypochondriacus, while lacked of relevant animal nutrition studies on Amaranthus hypochondriacus. Please add the updated references to the Introduction.
Question 3: In the Materials and Methods, in the line 148 to 150, the formulas should be presented in a format specific to “ANIMALS”; Then, the detection process requires simplified expression in line 158 to 160; All of the Table should be corrected to the correct format, like the decimal points (in table 1) and font size (p Value).
Question 4: The author should add the updated references from the SCIE related to the results of the study.
Some general comments
There are too much formal error in the manuscript, the author must revise carefully!
Question 5: This is lack of whitespace in Line 7 to 8, Line 45 to 46. There are many other places where the same error exists, you should You should check the full manuscript.
Question 6: there are the wrong format for reference. Many mistakes need to be corrected.
Author Response
请参阅附件

Reviewer 2 Report
Dear Authors:
The manuscript is well written but you need to work on references. The most important concern I have with the manuscript is the originality of the study. It is well recognized that plants' nutritional values go down, and so do degradation characteristics when plants mature. What was your hypothesis?
